# Higher-Dimensional Quantum Walk in Terms of Quantum Bernoulli Noises

**DOI:** 10.3390/e22050504

**Published:** 2020-04-28

**Authors:** Ce Wang, Caishi Wang

**Affiliations:** 1School of Mathematics and Statistics, Huazhong University of Science and Technology, Wuhan 430074, China; 2School of Mathematics and Statistics, Northwest Normal University, Lanzhou 730070, China; wangcs@nwnu.edu.cn

**Keywords:** quantum Bernoulli noises, quantum walk, quantum white noises, quantum probability, 81S25, 60G50, 81P68

## Abstract

As a discrete-time quantum walk model on the one-dimensional integer lattice Z, the quantum walk recently constructed by Wang and Ye [Caishi Wang and Xiaojuan Ye, Quantum walk in terms of quantum Bernoulli noises, Quantum Information Processing 15 (2016), 1897–1908] exhibits quite different features. In this paper, we extend this walk to a higher dimensional case. More precisely, for a general positive integer d≥2, by using quantum Bernoulli noises we introduce a model of discrete-time quantum walk on the *d*-dimensional integer lattice $Z^{d}$, which we call the *d*-dimensional QBN walk. The *d*-dimensional QBN walk shares the same coin space with the quantum walk constructed by Wang and Ye, although it is a higher dimensional extension of the latter. Moreover we prove that, for a range of choices of its initial state, the *d*-dimensional QBN walk has a limit probability distribution of *d*-dimensional standard Gauss type, which is in sharp contrast with the case of the usual higher dimensional quantum walks. Some other results are also obtained.

## 1. Introduction

As quantum analogs of classical random walks, quantum walks [1] have found wide application in quantum information, quantum computing and many other fields [2,3]. In the past two decades, quantum walks with a finite number of internal degrees of freedom have been intensively studied and many deep results have been obtained (see [2,3,4,5,6] and references therein). For example, Konno [5] found that a one-dimensional quantum walk with two internal degrees of freedom usually has a limit probability distribution with scaling speed *n*, instead of $\sqrt{n}$, which is far from being Gaussian.

Quantum Bernoulli noises refer to the family of annihilation and creation operators acting on Bernoulli functionals, which satisfy a canonical anti-commutation relation (CAR) in equal time, and can provide an approach to the effects of environment on an open quantum system [7,8]. In 2016, by using quantum Bernoulli noises, Wang and Ye [9] introduced a discrete-time quantum walk model on the one-dimensional integer lattice Z, which we call the one-dimensional QBN walk below.

Belonging to the category of unitary quantum walks, the one-dimensional QBN walk, however, exhibits quite different features. It takes the space H of square integrable Bernoulli functionals as its coin space, hence has infinitely many internal degrees of freedom since H is infinite-dimensional. Moreover, for some special choices of the initial state, it has the same limit probability distribution as the classical random walk [9], which is in marked contrast to the case of the usual unitary quantum walks (e.g., the Hadamard walk [5]). From a physical point of view [10], this behavior of the one-dimensional QBN walk might help understand the effects of decoherence in quantum walks.

Recent years have seen much attention paid to quantum walks on higher dimensional integer lattices. Mackay et al. [11] extended the Hadamard walk to a higher dimensional case and examined the time dependence of the standard deviation. Segawa and Konno [12] considered a quantum walk driven by many coins and found that the number of coins can have an important effect on the behavior of the walk. More recently, Komatsu and Konno [4] investigated stationary amplitudes of quantum walks on the higher-dimensional integer lattice. There are other works about quantum walks on higher dimensional integer lattices (see e.g., [13,14,15]).

In this paper, we would like to extend the one-dimensional QBN walk to a higher dimensional case. More precisely, for a general positive integer d≥2, we will use quantum Bernoulli noises to introduce a model of discrete-time quantum walk on the *d*-dimensional integer lattice $Z^{d}$. Our main work is as follows.

For each n≥0, by using quantum Bernoulli noises, we construct $2^{d}$ self-adjoint operators $C_{n}^{\left(\varepsilon \right)}$, $\varepsilon \in {\left\{-1,+1\right\}}^{d}$, which act on the space H of square integrable Bernoulli functionals. We prove that $C_{n}^{\left(\varepsilon \right)}$, $\varepsilon \in {\left\{-1,+1\right\}}^{d}$ are pairwise orthogonal and moreover their sum is unitary.By taking the operators $C_{n}^{\left(\varepsilon \right)}$, $\varepsilon \in {\left\{-1,+1\right\}}^{d}$, n≥0 as coin operators, we establish a model of discrete-time quantum walk on $Z^{d}$, which we call the *d*-dimensional QBN walk. Of this walk, we obtain a unitary representation in the function space $l^{2}\left(Z^{d},H\right)$ and a characterization in the tensor space $l^{2}\left(Z^{d}\right)⊗H$.Under some mild conditions, we obtain a link between amplitudes of the *d*-dimensional QBN walk and those of the one-dimensional QBN walk. And based on this link, we find that, for a range of choices of its initial state, the *d*-dimensional QBN walk has a limit probability distribution of *d*-dimensional standard Gauss type.

As is seen, the coin space of the *d*-dimensional QBN walk is just the space H of square integrable Bernoulli functionals, which is infinite-dimensional. Thus the *d*-dimensional QBN walk has infinitely many internal degrees of freedom. It should be also mentioned that the *d*-dimensional QBN walk shares the same coin space with the one-dimensional QBN walk, although it is a higher dimensional extension of the latter.

This paper consists of five sections. In Section 2, we briefly recall some necessary notions and facts about quantum Bernoulli noises. Our main work then lies in Section 3 and Section 4. Here, among others, we prove several supporting theorems, define our quantum walk model and examine its fundamental properties. Finally in Section 5, we make some conclusion remarks.

## 2. Preliminaries

In this section, we briefly recall some necessary notions and facts about quantum Bernoulli noises. We refer to [7] for details about quantum Bernoulli noises.

Throughout this paper, Z always denotes the set of all integers, while N means the set of all nonnegative integers. We denote by Γ the finite power set of N, namely
(1)Γ={σ∣σ⊂Nand#σ<∞},
where #σ means the cardinality of σ. Unless otherwise stated, letters like *j*, *k* and *n* stand for nonnegative integers, namely elements of N.

Let Ω be the set of all mappings ω:N↦{−1,1}, and ${\left(\zeta_{n}\right)}_{n\geq 0}$ the sequence of canonical projections on Ω given by
(2)$$\zeta_{n}\left(\omega \right)=\omega \left(n\right),\;\omega \in \Omega .$$

Let F be the σ-field on Ω generated by the sequence ${\left(\zeta_{n}\right)}_{n\geq 0}$, and ${\left(p_{n}\right)}_{n\geq 0}$ a given sequence of positive numbers with the property that $0<p_{n}<1$ for all n≥0. Then there exists a unique probability measure P on F such that
(3)$$P\circ {\left(\zeta_{n_{1}},\zeta_{n_{2}},\cdots ,\zeta_{n_{k}}\right)}^{-1}\left\{(\epsilon_{1},\epsilon_{2},\cdots ,\epsilon_{k})\right\}=\prod_{j=1}^{k}p_{j}^{\frac{1+\epsilon_{j}}{2}}{\left(1-p_{j}\right)}^{\frac{1-\epsilon_{j}}{2}}$$
for $n_{j}\in N$, $\epsilon_{j}\in \left\{-1,1\right\}$ (1≤j≤k) with $n_{i}\neq n_{j}$ when i≠j and k∈N with k≥1. Thus one has a probability measure space (Ω,F,P), which is referred to as the Bernoulli space and random variables on it are known as Bernoulli functionals.

Let $Z={\left(Z_{n}\right)}_{n\geq 0}$ be the sequence of Bernoulli functionals generated by sequence ${\left(\zeta_{n}\right)}_{n\geq 0}$, namely
(4)$$Z_{n}=\frac{\zeta_{n}+q_{n}-p_{n}}{2\sqrt{p_{n}q_{n}}},\;n\geq 0,$$
where $q_{n}=1-p_{n}$. Clearly $Z={\left(Z_{n}\right)}_{n\geq 0}$ is an independent sequence of random variables on the probability measure space (Ω,F,P). Let H be the space of square integrable complex-valued Bernoulli functionals, namely
(5)$$H=L^{2}\left(\Omega ,F,P\right).$$
We denote by 〈·,·〉 the usual inner product of the space H, and by ∥·∥ the corresponding norm. It is known that *Z* has the chaotic representation property. Thus $Z=\{Z_{\sigma }|\sigma \in \Gamma \}$ form an orthonormal basis (ONB) of H, which is known as the canonical ONB of H. Here $Z_{\emptyset }=1$ and
(6)$$Z_{\sigma }=\prod_{j\in \sigma }Z_{j},\;\sigma \in \Gamma ,\;\sigma \neq \emptyset .$$

Clearly H is infinite-dimensional as a complex Hilbert space.

**Lemma** **1.**([7]) *For each k∈N, there exists a bounded operator $\partial_{k}$ on H such that*
(7)$$\partial_{k}Z_{\sigma }=1_{\sigma }\left(k\right)Z_{\sigma \setminus k},\;\partial_{k}^{*}Z_{\sigma }=\left[1-1_{\sigma }\left(k\right)\right]Z_{\sigma \cup k}\;\sigma \in \Gamma ,\sigma \in \Gamma ,$$
*where $\partial_{k}^{*}$ denotes the adjoint of $\partial_{k}$, σ∖k=σ∖{k}, σ∪k=σ∪{k} and $1_{\sigma }\left(k\right)$ the indicator of σ as a subset of N.*

The operators $\partial_{k}$ and $\partial_{k}^{*}$ are usually known as the annihilation and creation operators acting on Bernoulli functionals, respectively. And the family ${\left\{\partial_{k},\partial_{k}^{*}\right\}}_{k\geq 0}$ is referred to as quantum Bernoulli noises. The next lemma shows that quantum Bernoulli noises satisfy the canonical anti-commutation relations (CAR) in equal-time.

**Lemma** **2.**([7]) *Let k, l∈N. Then it holds true that*
(8)$$\partial_{k}\partial_{l}=\partial_{l}\partial_{k},\;\partial_{k}^{*}\partial_{l}^{*}=\partial_{l}^{*}\partial_{k}^{*},\;\partial_{k}^{*}\partial_{l}=\partial_{l}\partial_{k}^{*}\;\left(k\neq l\right)$$
*and*
(9)$$\partial_{k}\partial_{k}=\partial_{k}^{*}\partial_{k}^{*}=0,\;\partial_{k}\partial_{k}^{*}+\partial_{k}^{*}\partial_{k}=I,$$
*where I is the identity operator on H.*

For a nonnegative integer n≥0, one can define, respectively, two self-adjoint operators $L_{n}$ and $R_{n}$ on H in the following manner
(10)$$L_{n}=\frac{1}{2}\left(\partial_{n}^{*}+\partial_{n}-I\right),\;R_{n}=\frac{1}{2}\left(\partial_{n}^{*}+\partial_{n}+I\right),$$
where *I* is the identity operator on H. It then follows from Lemma 2 that the operators $L_{n}$, $R_{n}$, n≥0, form a commutative family, namely
(11)$$L_{k}L_{l}=L_{l}L_{k},\;R_{k}L_{l}=L_{l}R_{k},\;R_{k}R_{l}=R_{l}R_{k},\;k,\;l\geq 0.$$

**Lemma** **3.**([9]) *For all n≥0, $R_{n}+L_{n}$ is a unitary operator on H and moreover it holds that*
(12)$$R_{n}^{2}=R_{n},\;R_{n}L_{n}=L_{n}R_{n}=0,\;L_{n}^{2}=-L_{n}.$$

In view of the commutativity of family $\left\{L_{n},\;R_{n}|n\geq 0\right\}$, we can naturally introduce the following symbols
(13)$$L_{\sigma }=\prod_{k\in \sigma }L_{k},\;\sigma \neq \emptyset ,\;\sigma \in \Gamma$$
and $L_{\emptyset }=I$, the identity operator on H. Similarly we can define $R_{\sigma }$ for any σ∈Γ. It can be verified that $L_{\sigma }$, $R_{\sigma }$, σ∈Γ also form a commutative family of self-adjoint operators on H. Additionally, it can be shown that $L_{\sigma }R_{\tau }=0$ whenever σ, τ∈Γ with σ∩τ≠∅.

## 3. Definition and Fundamental Properties

In this section, we prove some supporting theorems, present the definition of our quantum walk and examine its fundamental properties.

In what follows, we always assume that d≥2 is a given positive integer and Λ={−1,+1}. We denote by $\Lambda^{d}$ the *d*-fold cartesian product of Λ, and by $H^{⊗d}$ the *d*-fold tensor product space of H. In addition, we assume that $K:H^{⊗d}\to H$ is a fixed unitary isomorphism. Such a unitary isomorphism exists because H is infinite-dimensional and separable.

### 3.1. Coin Operators

This subsection constructs our coin operators, which will play a fundamental role in defining our quantum walk.

Recall that $L_{n}=\frac{1}{2}\left(\partial_{n}^{*}+\partial_{n}-I\right)$, $R_{n}=\frac{1}{2}\left(\partial_{n}^{*}+\partial_{n}+I\right)$ for n≥0. In what follows, for notational convenience we rewrite $B_{n}^{\left(-1\right)}=L_{n}$, $B_{n}^{\left(+1\right)}=R_{n}$. And for $\varepsilon =\left(\varepsilon_{1},\varepsilon_{2},\cdots ,\varepsilon_{d}\right)\in \Lambda^{d}$, we use the symbol
(14)$$\overset{d}{\underset{i=1}{⨂}}B_{n}^{\left(\varepsilon_{i}\right)}=B_{n}^{\left(\varepsilon_{1}\right)}⊗B_{n}^{\left(\varepsilon_{2}\right)}⊗\cdots ⊗B_{n}^{\left(\varepsilon_{d}\right)}$$
to mean the tensor product of $B_{n}^{\left(\varepsilon_{1}\right)}$, $B_{n}^{\left(\varepsilon_{2}\right)}$, ⋯, $B_{n}^{\left(\varepsilon_{d}\right)}$. Clearly, ${⨂}_{i=1}^{d}B_{n}^{\left(\varepsilon_{i}\right)}$ is a bounded operator on $H^{⊗d}$ for each $\varepsilon =\left(\varepsilon_{1},\varepsilon_{2},\cdots ,\varepsilon_{d}\right)\in \Lambda^{d}$.

**Definition** **1.**
*For n≥0 and $\varepsilon =\left(\varepsilon_{1},\varepsilon_{2},\cdots ,\varepsilon_{d}\right)\in \Lambda^{d}$, we define an operator $C_{n}^{\left(\varepsilon \right)}$ on H as*
(15)$$C_{n}^{\left(\varepsilon \right)}=K\left(\overset{d}{\underset{i=1}{⨂}}B_{n}^{\left(\varepsilon_{i}\right)}\right)K^{-1},$$
*where, as indicated above, K is the unitary isomorphism from $H^{⊗d}$ to H and $K^{-1}$ is the inverse of K.*


**Theorem** **1.**
*Let n≥0. Then $C_{n}^{\left(\varepsilon \right)}$, $\varepsilon \in \Lambda^{d}$ are self-adjoint operators on H. And moreover they admit the following operation properties:*
*(i)* 
*$C_{n}^{\left(\varepsilon \right)}C_{n}^{\left(\varepsilon^{\prime }\right)}=0$, $\varepsilon \neq \varepsilon^{\prime }$, ε, $\varepsilon^{\prime }\in \Lambda^{d}$;*
*(ii)* 
*$\sum_{\varepsilon \in \Lambda^{d}}C_{n}^{\left(\varepsilon \right)}$ is a unitary operator on H.*



**Proof.** For each $\varepsilon =\left(\varepsilon_{1},\varepsilon_{2},\cdots ,\varepsilon_{d}\right)\in \Lambda^{d}$, it follows from the fact of $B_{n}^{\left(\varepsilon_{i}\right)}$ being self-adjoint for all i∈{1,2,⋯,d} that ${⨂}_{i=1}^{d}B_{n}^{\left(\varepsilon_{i}\right)}$ is a self-adjoint operator on $H^{⊗d}$, which, together with the fact of K being unitary, implies that the operator defined by (Equation 15), namely $C_{n}^{\left(\varepsilon \right)}$, is self-adjoint as an operator on H.Let ε, $\varepsilon^{\prime }\in \Lambda^{d}$ with $\varepsilon \neq \varepsilon^{\prime }$. Then there is some j∈{1,2,⋯,d} such that $\varepsilon_{j}\neq \varepsilon_{j}^{\prime }$, where $\varepsilon_{j}$ and $\varepsilon_{j}^{\prime }$ are the *j*th components of ε and $\varepsilon^{\prime }$, respectively. By Lemma 3, $B_{n}^{\left(\varepsilon_{j}\right)}B_{n}^{\left(\varepsilon_{j}^{\prime }\right)}=0$, which implies that ${⨂}_{i=1}^{d}\left(B_{n}^{\left(\varepsilon_{i}\right)}B_{n}^{\left(\varepsilon_{i}^{\prime }\right)}\right)=0$. Thus, we have
$$C_{n}^{\left(\varepsilon \right)}C_{n}^{\left(\varepsilon^{\prime }\right)}=K\left(\overset{d}{\underset{i=1}{⨂}}\left(B_{n}^{\left(\varepsilon_{i}\right)}B_{n}^{\left(\varepsilon_{i}^{\prime }\right)}\right)\right)K^{-1}=0.$$This completes the proof of property (i).Next, we verify property (ii). In fact, for each i∈{1,2,⋯,d}, it follows from Lemma 3 that
$$\sum_{\varepsilon_{i}\in \Lambda }B_{n}^{\left(\varepsilon_{i}\right)}=B_{n}^{\left(+1\right)}+B_{n}^{\left(-1\right)}=R_{n}+L_{n}$$
is a unitary operator on H. Thus, by the property of operator tensor product, we know that
$$\sum_{\varepsilon \in \Lambda^{d}}\left(\overset{d}{\underset{i=1}{⨂}}B_{n}^{\left(\varepsilon_{i}\right)}\right)=\overset{d}{\underset{i=1}{⨂}}\left(\sum_{\varepsilon_{i}\in \Lambda }B_{n}^{\left(\varepsilon_{i}\right)}\right)$$
is a unitary operator on $H^{⊗d}$, which, together with fact that $K:H^{⊗d}\to H$ is a unitary isomorphism, implies that
$$\sum_{\varepsilon \in \Lambda^{d}}C_{n}^{\left(\varepsilon \right)}=\sum_{\varepsilon \in \Lambda^{d}}K\left(\overset{d}{\underset{i=1}{⨂}}B_{n}^{\left(\varepsilon_{i}\right)}\right)K^{-1}=K\left[\sum_{\varepsilon \in \Lambda^{d}}\left(\overset{d}{\underset{i=1}{⨂}}B_{n}^{\left(\varepsilon_{i}\right)}\right)\right]K^{-1}$$
is a unitary operator on H. □

### 3.2. Definition and Unitary Representation

In this subsection, we present the definition of our quantum walk and find out its unitary representation. As usual, we set $Z^{d}$ to be the *d*-fold cartesian product of Z, and we denote by $l^{2}\left(Z^{d},H\right)$ the space of square summable functions defined on $Z^{d}$ and valued in H, namely
(16)$$l^{2}\left(Z^{d},H\right)=\left\{\;W:Z^{d}\to H|\sum_{x\in Z^{d}}{∥W\left(x\right)∥}^{2}<\infty \;\right\},$$
where ∥·∥ means the norm in H. As is known, $l^{2}\left(Z^{d},H\right)$ is a separable Hilbert space with the inner product given by
(17)$${\langle U,V\rangle }_{l^{2}\left(Z^{d},H\right)}=\sum_{x\in Z^{d}}\langle U(x),V(x)\rangle ,\;U,\;V\in l^{2}\left(Z^{d},H\right),$$
where 〈·,·〉 is the inner product in H. By convention, elements of $l^{2}\left(Z^{d},H\right)$ are usually known as vectors. A vector $W\in l^{2}\left(Z^{d},H\right)$ is called a unit vector if ${∥W∥}_{l^{2}\left(Z^{d},H\right)}=1$, where ${∥\cdot ∥}_{l^{2}\left(Z^{d},H\right)}$ stands for the norm in $l^{2}\left(Z^{d},H\right)$. Note that each unit vector $W\in l^{2}\left(Z^{d},H\right)$ makes a probability distribution $x\mapsto {∥W\left(x\right)∥}^{2}$ on $Z^{d}$.

**Theorem** **2.**
*Let n≥0. Then, for each $W\in l^{2}\left(Z^{d},H\right)$, both the function $W^{\prime }:Z^{d}\to H$ defined by*
(18)$$W^{\prime }\left(x\right)=\sum_{\varepsilon \in \Lambda^{d}}C_{n}^{\left(\varepsilon \right)}W\left(x-\varepsilon \right),\;x\in Z^{d}$$
*and the function $W^{\prime \prime }:Z^{d}\to H$ defined by*
(19)$$W^{\prime \prime }\left(x\right)=\sum_{\varepsilon \in \Lambda^{d}}C_{n}^{\left(\varepsilon \right)}W\left(x+\varepsilon \right),\;x\in Z^{d}$$
*all belong to $l^{2}\left(Z^{d},H\right)$, and moreover $∥W^{\prime }{∥}_{l^{2}\left(Z^{d},H\right)}=∥W^{\prime \prime }{∥}_{l^{2}\left(Z^{d},H\right)}={∥W∥}_{l^{2}\left(Z^{d},H\right)}$.*


**Proof.** By using Theorem 1, we have
$$\sum_{x\in Z^{d}}{∥W^{\prime }\left(x\right)∥}^{2}=\sum_{x\in Z^{d}}∥\sum_{\varepsilon \in \Lambda^{d}}C_{n}^{\left(\varepsilon \right)}W\left(x-\varepsilon \right){∥}^{2}=\sum_{x\in Z^{d}}\sum_{\varepsilon \in \Lambda^{d}}∥C_{n}^{\left(\varepsilon \right)}W\left(x-\varepsilon \right){∥}^{2},$$
which together with the invariance
$$\sum_{x\in Z^{d}}\sum_{\varepsilon \in \Lambda^{d}}∥C_{n}^{\left(\varepsilon \right)}W\left(x-\varepsilon \right){∥}^{2}=\sum_{x\in Z^{d}}\sum_{\varepsilon \in \Lambda^{d}}∥C_{n}^{\left(\varepsilon \right)}W\left(x\right){∥}^{2}$$
gives
$$\sum_{x\in Z^{d}}{∥W^{\prime }\left(x\right)∥}^{2}=\sum_{x\in Z^{d}}\sum_{\varepsilon \in \Lambda^{d}}∥C_{n}^{\left(\varepsilon \right)}W\left(x\right){∥}^{2}=\sum_{x\in Z^{d}}∥\sum_{\varepsilon \in \Lambda^{d}}C_{n}^{\left(\varepsilon \right)}W\left(x\right){∥}^{2},$$
which, together with the fact that $\sum_{\varepsilon \in \Lambda^{d}}C_{n}^{\left(\varepsilon \right)}$ is a unitary operator on H, implies that
$$\sum_{x\in Z^{d}}∥W^{\prime }{\left(x\right)∥}^{2}=\sum_{x\in Z^{d}}{∥W\left(x\right)∥}^{2}.$$Therefore $W^{\prime }\in l^{2}\left(Z^{d},H\right)$ and $∥W^{\prime }{∥}_{l^{2}\left(Z^{d},H\right)}={∥W∥}_{l^{2}\left(Z^{d},H\right)}$. Similarly, we can show that $W^{\prime \prime }\in l^{2}\left(Z^{d},H\right)$ and $∥W^{\prime \prime }{∥}_{l^{2}\left(Z^{d},H\right)}={∥W∥}_{l^{2}\left(Z^{d},H\right)}$. □

Based on Theorem 2, we can now present the definition of our quantum walk on $Z^{d}$ as follows.

**Definition** **2.**
*The d-dimensional QBN walk is a discrete-time quantum walk on the d-dimensional integer latice $Z^{d}$ that satisfies the following requirements.*

*The walk takes $l^{2}\left(Z^{d},H\right)$ as its state space, and its states are represented by unit vectors in $l^{2}\left(Z^{d},H\right)$.*

*The time evolution of the walk is governed by equation*
(20)$$W_{n+1}\left(x\right)=\sum_{\varepsilon \in \Lambda^{d}}C_{n}^{\left(\varepsilon \right)}W_{n}\left(x-\varepsilon \right),\;x\in Z^{d},\;n\geq 0,$$
*where $W_{n}$ denotes the state of the walk at time n≥0, and in particular $W_{0}$ is the initial state of the walk.*


*In that case, the function $x\mapsto ∥W_{n}{\left(x\right)∥}^{2}$ on $Z^{d}$ is called the probability distribution of the walk at time n≥0, while the quantity $∥W_{n}{\left(x\right)∥}^{2}$ is the probability to find out the walker at position $x\in Z^{d}$ and time n≥0.*


It is well known that $l^{2}\left(Z^{d},H\right)≅l^{2}\left(Z^{d}\right)⊗H$. This just means that $l^{2}\left(Z^{d}\right)$ describes the position of the *d*-dimensional QBN walk, while H describes its internal degrees of freedom. As usual, H is called the coin space of the walk. Clearly, the *d*-dimensional QBN walk has infinitely many internal degrees of freedom because its coin space H is infinite-dimensional.

**Theorem** **3.**
*For each n≥0, there exists a unitary operator $U_{n}^{\left(d\right)}$ on $l^{2}\left(Z^{d},H\right)$ such that*
(21)$$\left(U_{n}^{\left(d\right)}W\right)\left(x\right)=\sum_{\varepsilon \in \Lambda^{d}}C_{n}^{\left(\varepsilon \right)}W\left(x-\varepsilon \right),\;x\in Z^{d},\;W\in l^{2}\left(Z^{d},H\right)$$
*and*
(22)$$\left[{\left(U_{n}^{\left(d\right)}\right)}^{*}W\right]\left(x\right)=\sum_{\varepsilon \in \Lambda^{d}}C_{n}^{\left(\varepsilon \right)}W\left(x+\varepsilon \right),\;x\in Z^{d},\;W\in l^{2}\left(Z^{d},H\right),$$
*where ${\left(U_{n}^{\left(d\right)}\right)}^{*}$ denotes the adjoint of $U_{n}^{\left(d\right)}$.*


**Proof.** For each $W\in l^{2}\left(Z^{d},H\right)$, denote by $W^{\prime }$ the function given by
$$W^{\prime }\left(x\right)=\sum_{\varepsilon \in \Lambda^{d}}C_{n}^{\left(\varepsilon \right)}W\left(x-\varepsilon \right),\;x\in Z^{d},$$
which, by Theorem 2, belongs to $l^{2}\left(Z^{d},H\right)$. Thus, we can define an operator $U_{n}^{\left(d\right)}$ on $l^{2}\left(Z^{d},H\right)$ in the following manner
(23)$$U_{n}^{\left(d\right)}W=W^{\prime },\;W\in l^{2}\left(Z^{d},H\right).$$It is easy to see that $U_{n}^{\left(d\right)}$ is linear. And moreover, by Theorem 2, we know that $U_{n}^{\left(d\right)}$ is even an isometry, which means that $U_{n}^{\left(d\right)}$ has an adjoint ${\left(U_{n}^{\left(d\right)}\right)}^{*}$.Let *U*, $V\in l^{2}\left(Z^{d},H\right)$. Then, by general properties of the adjoint of an operator, we have
$${\langle {\left(U_{n}^{\left(d\right)}\right)}^{*}U,V\rangle }_{l^{2}\left(Z^{d},H\right)}={\langle U,U_{n}^{\left(d\right)}V\rangle }_{l^{2}\left(Z^{d},H\right)}=\sum_{x\in Z^{d}}\sum_{\varepsilon \in \Lambda^{d}}\langle U\left(x\right),C_{n}^{\left(\varepsilon \right)}V\left(x-\varepsilon \right)\rangle ,$$
which, together with the fact of $C_{n}^{\left(\varepsilon \right)}$ being self-adjoint, gives
$$\begin{array}{cc} {\langle {\left(U_{n}^{\left(d\right)}\right)}^{*}U,V\rangle }_{l^{2}\left(Z^{d},H\right)}\; & =\sum_{x\in Z^{d}}\sum_{\varepsilon \in \Lambda^{d}}\langle C_{n}^{\left(\varepsilon \right)}U\left(x\right),V\left(x-\varepsilon \right)\rangle \\ \; & =\sum_{x\in Z^{d}}\sum_{\varepsilon \in \Lambda^{d}}\langle C_{n}^{\left(\varepsilon \right)}U\left(x+\varepsilon \right),V\left(x\right)\rangle \\ \; & =\sum_{x\in Z^{d}}\langle U^{\prime \prime }\left(x\right),V\left(x\right)\rangle \\ \; & ={\langle U^{\prime \prime },V\rangle }_{l^{2}\left(Z^{d},H\right)}, \end{array}$$
where $U^{\prime \prime }$ is the function given by
$$U^{\prime \prime }\left(x\right)=\sum_{\varepsilon \in \Lambda^{d}}C_{n}^{\left(\varepsilon \right)}U\left(x+\varepsilon \right),\;x\in Z^{d}.$$It then follows from the arbitrariness of $V\in l^{2}\left(Z^{d},H\right)$ that $(U_{n}^{\left(d\right)}{)}^{*}U=U^{\prime \prime }$, namely
$$\left[{\left(U_{n}^{\left(d\right)}\right)}^{*}U\right]\left(x\right)=\sum_{\varepsilon \in \Lambda^{d}}C_{n}^{\left(\varepsilon \right)}U\left(x+\varepsilon \right),\;x\in Z^{d}.$$This shows that (Equation 22) holds. A direct calculation yields that
$${\left(U_{n}^{\left(d\right)}\right)}^{*}U_{n}^{\left(d\right)}=U_{n}^{\left(d\right)}{\left(U_{n}^{\left(d\right)}\right)}^{*}=I,$$
where I means the identity operator on $l^{2}\left(Z^{d},H\right)$. Therefore, $U_{n}^{\left(d\right)}$ is a unitary operator satisfying (Equation 21) and (Equation 22). □

Applying Theorem 3 to Definition 2, we come to the next theorem, which shows that the *d*-dimensional QBN walk belongs to the category of unitary quantum walks.

**Theorem** **4.**
*Let $W_{n}$ be the state of the d-dimensional QBN walk at time n≥0. Then*
(24)$$W_{n}=\left(\prod_{k=0}^{n-1}U_{k}^{\left(d\right)}\right)W_{0},\;n\geq 1,$$
*where $W_{0}$ is the initial state of the walk.*


### 3.3. Characterization in Tensor Space

As is seen, the *d*-dimensional QBN walk is formulated in the function space $l^{2}\left(Z^{d},H\right)$. In the present subsection, we reformulate it in the tensor space $l^{2}\left(Z^{d}\right)⊗H$, which is isomorphic to $l^{2}\left(Z^{d},H\right)$ in the sense of unitary isomorphism.

Let $J:l^{2}\left(Z^{d},H\right)\to l^{2}\left(Z^{d}\right)⊗H$ be the canonical unitary isomorphism. Then, J satisfies that
(25)$$JW_{f,\xi }=f⊗\xi ,\;f\in l^{2}\left(Z^{d}\right),\;\xi \in H,$$
where $W_{f,\xi }$ is the function defined by $W_{f,\xi }\left(x\right)=f\left(x\right)\xi$, $x\in Z^{d}$. As is indicated in Theorem 4, unitary operator sequence ${\left(U_{n}^{\left(d\right)}\right)}_{n\geq 0}$ plays an important role in describing the *d*-dimensional QBN walk.

In the following, for each n≥0, we denote by $V_{n}^{\left(d\right)}$ the counterpart of $U_{n}^{\left(d\right)}$ in tensor space $l^{2}\left(Z^{d}\right)⊗H$, namely
(26)$$V_{n}^{\left(d\right)}=J\;U_{n}^{\left(d\right)}J^{-1}.$$
Then ${\left(V_{n}^{\left(d\right)}\right)}_{n\geq 0}$ is a sequence of unitary operators on $l^{2}\left(Z^{d}\right)⊗H$. Thus, from a physical point of view, we naturally come to the next observation.

**Remark** **1.**
*The d-dimensional QBN walk can be viewed as a unitary evolution determined by the unitary operator sequence ${\left(V_{n}^{\left(d\right)}\right)}_{n\geq 0}$ on the tensor space $l^{2}\left(Z^{d}\right)⊗H$.*


We now consider the structure of unitary operators $V_{n}^{\left(d\right)}$, n≥0. Let $f\in l^{2}\left(Z^{d}\right)$ and ξ∈H. Then, for $\varepsilon \in \Lambda^{d}$, by letting $g^{\left(\varepsilon \right)}\left(x\right)=f\left(x-\varepsilon \right)$, $x\in Z^{d}$, we have
(27)$$g^{\left(\varepsilon \right)}=\sum_{x\in Z^{d}}|\delta_{x}\rangle \langle \delta_{x-\varepsilon }|f,$$
where the series on the righthand side converges in the norm of $l^{2}\left(Z^{d}\right)$. By Theorem 3, we have
$$\left[U_{n}^{\left(d\right)}W_{f,\xi }\right]\left(x\right)=\sum_{\varepsilon \in \Lambda^{d}}C_{n}^{\left(\varepsilon \right)}W_{f,\xi }\left(x-\varepsilon \right)=\sum_{\varepsilon \in \Lambda^{d}}C_{n}^{\left(\varepsilon \right)}\left[g^{\left(\varepsilon \right)}\left(x\right)\xi \right]=\sum_{\varepsilon \in \Lambda^{d}}g^{\left(\varepsilon \right)}\left(x\right)C_{n}^{\left(\varepsilon \right)}\xi ,\;x\in Z^{d},$$
which implies that
$$\left[U_{n}^{\left(d\right)}W_{f,\xi }\right]\left(x\right)=\sum_{\varepsilon \in \Lambda^{d}}W_{g^{\left(\varepsilon \right)},C_{n}^{\left(\varepsilon \right)}\xi }\left(x\right),\;x\in Z^{d}.$$

Thus, as vectors in $l^{2}\left(Z^{d},H\right)$, we have
$$U_{n}^{\left(d\right)}W_{f,\xi }=\sum_{\varepsilon \in \Lambda^{d}}W_{g^{\left(\varepsilon \right)},C_{n}^{\left(\varepsilon \right)}\xi },$$
which, together with J and $J^{-1}$, yields
$$V_{n}^{\left(d\right)}\left(f⊗\xi \right)=\left[J\;U_{n}^{\left(d\right)}J^{-1}\right]\left(f⊗\xi \right)=\left[J\;U_{n}^{\left(d\right)}\right]W_{f,\xi }=J\sum_{\varepsilon \in \Lambda^{d}}W_{g^{\left(\varepsilon \right)},C_{n}^{\left(\varepsilon \right)}\xi }=\sum_{\varepsilon \in \Lambda^{d}}g^{\left(\varepsilon \right)}⊗C_{n}^{\left(\varepsilon \right)}\xi ,$$
which together with (Equation 27) implies that
$$V_{n}^{\left(d\right)}\left(f⊗\xi \right)=\left[\sum_{x\in Z^{d}}\sum_{\varepsilon \in \Lambda^{d}}|\delta_{x}\rangle \langle \delta_{x-\varepsilon }|⊗C_{n}^{\left(\varepsilon \right)}\right]\left(f⊗\xi \right).$$

Therefore, by the arbitrariness of choosing $f\in l^{2}\left(Z^{d}\right)$ and ξ∈H, we come to the next result, which actually offers a characterization of the *d*-dimensional QBN walk in tensor space.

**Theorem** **5.**
*Let n≥0. Then, the unitary operator $V_{n}^{\left(d\right)}$ has a structure of the following form*
(28)$$V_{n}^{\left(d\right)}=\sum_{x\in Z^{d}}\sum_{\varepsilon \in \Lambda^{d}}|\delta_{x}\rangle \langle \delta_{x-\varepsilon }|⊗C_{n}^{\left(\varepsilon \right)},$$
*where $\left\{\delta_{x}|x\in Z^{d}\right\}$ is the canonical ONB of $l^{2}\left(Z^{d}\right)$ and $|\delta_{x}\rangle \langle \delta_{x-\varepsilon }|$ is the Dirac operator.*


## 4. Limit Probability Distribution

In the present section, we focus on exploring limit probability distribution of the *d*-dimensional QBN walk. To be convenient, we additionally denote by $l^{2}\left(Z,H\right)$ the space of square summable functions defined on Z and valued in H.

### 4.1. Amplitude Formula

**Lemma** **4.**([9]) *For each n≥0, there exists a unitary operator $U_{n}$ on $l^{2}\left(Z,H\right)$ such that*
(29)$$\left(U_{n}\Phi \right)\left(z\right)=R_{n}\Phi \left(z-1\right)+L_{n}\Phi \left(z+1\right),\;z\in Z,\;\Phi \in l^{2}\left(Z,H\right)$$
*and*
(30)$$\left(U_{n}^{*}\Phi \right)\left(z\right)=R_{n}\Phi \left(z+1\right)+L_{n}\Phi \left(z-1\right),\;z\in Z,\;\Phi \in l^{2}\left(Z,H\right),$$
*where $U_{n}^{*}$ stands for the adjoint of $U_{n}$.*

For vectors $\varphi^{\left(1\right)}$, $\varphi^{\left(2\right)}$, ⋯, $\varphi^{\left(d\right)}\in l^{2}\left(Z,H\right)$, it can be verified that the function defined by
$$x=\left(x_{1},x_{2},\cdots ,x_{d}\right)\;⟼\;K\left(\overset{d}{\underset{i=1}{⨂}}\varphi^{\left(i\right)}\left(x_{i}\right)\right)$$
belongs to $l^{2}\left(Z^{d},H\right)$. Moreover, this function even becomes a unit vector in $l^{2}\left(Z^{d},H\right)$ whenever $\varphi^{\left(1\right)}$, $\varphi^{\left(2\right)}$, ⋯, $\varphi^{\left(d\right)}$ are unit vectors in $l^{2}\left(Z,H\right)$.

As is shown above, $U_{n}$, n≥0 are unitary operators on $l^{2}\left(Z,H\right)$. Thus, for all unit vector $\varphi \in l^{2}\left(Z,H\right)$, vectors
$$\left(\prod_{k=0}^{n}U_{k}\right)\varphi ,\;n\geq 0$$
obviously make a sequence of unit vectors in $l^{2}\left(Z,H\right)$.

**Theorem** **6.**
*Let $\varphi^{\left(1\right)}$, $\varphi^{\left(2\right)}$, ⋯, $\varphi^{\left(d\right)}\in l^{2}\left(Z,H\right)$ be unit vectors in $l^{2}\left(Z,H\right)$. Suppose that the initial state $W_{0}$ of the d-dimensional QBN walk takes the following form*
(31)$$W_{0}\left(x\right)=K\left(\overset{d}{\underset{i=1}{⨂}}\varphi^{\left(i\right)}\left(x_{i}\right)\right),\;x=\left(x_{1},x_{2},\cdots ,x_{d}\right)\in Z^{d}.$$

*Then, for all n≥1, the state $W_{n}$ of the d-dimensional QBN walk at time n satisfies the following relation*
(32)$$W_{n}\left(x\right)=K\left(\overset{d}{\underset{i=1}{⨂}}\Phi_{n}^{\left(i\right)}\left(x_{i}\right)\right),\;x=\left(x_{1},x_{2},\cdots ,x_{d}\right)\in Z^{d},$$
*where*
(33)$$\Phi_{n}^{\left(i\right)}=\left(\prod_{k=0}^{n-1}U_{k}\right)\Phi_{0}^{\left(i\right)}$$
*with $\Phi_{0}^{\left(i\right)}=\varphi^{\left(i\right)}$ for all index i∈{1,2,⋯,d}.*


**Proof.** By Lemma 4, for all n≥0 and i∈{1,2,⋯,d}, $\Phi_{n}^{\left(i\right)}$ is a unit vector in $l^{2}\left(Z,H\right)$. Now, for each nonnegative integer n≥0, we define a function $W_{n}^{\prime }:Z^{d}\to H$ as
(34)$$W_{n}^{\prime }\left(x\right)=K\left(\overset{d}{\underset{i=1}{⨂}}\Phi_{n}^{\left(i\right)}\left(x_{i}\right)\right),\;x=\left(x_{1},x_{2},\cdots ,x_{d}\right)\in Z^{d}.$$Then, as indicated above, $W_{n}^{\prime }$, n≥0, are unit vectors in $l^{2}\left(Z^{d},H\right)$, and in particular
$$W_{0}^{\prime }\left(x\right)=K\left(\overset{d}{\underset{i=1}{⨂}}\Phi_{0}^{\left(i\right)}\left(x_{i}\right)\right)=K\left(\overset{d}{\underset{i=1}{⨂}}\varphi^{\left(i\right)}\left(x_{i}\right)\right)=W_{0}\left(x\right),\;x=\left(x_{1},x_{2},\cdots ,x_{d}\right)\in Z^{d},$$
which implies that $W_{0}^{\prime }=W_{0}$.On the other hand, for all n≥1 and i∈{1,2,⋯,d}, by using (Equation 33) and Lemma 4, we find that
$$\Phi_{n}^{\left(i\right)}\left(x_{i}\right)=R_{n-1}\Phi_{n-1}^{\left(i\right)}\left(x_{i}-1\right)+L_{n-1}\Phi_{n-1}^{\left(i\right)}\left(x_{i}+1\right),\;x_{i}\in Z,$$
which, together with the notation $B_{n-1}^{\left(+1\right)}=R_{n-1}$ and $B_{n-1}^{\left(-1\right)}=L_{n-1}$ (see Section 3.1 for details), gives
$$\Phi_{n}^{\left(i\right)}\left(x_{i}\right)=B_{n-1}^{\left(+1\right)}\Phi_{n-1}^{\left(i\right)}\left(x_{i}-1\right)+B_{n-1}^{\left(-1\right)}\Phi_{n-1}^{\left(i\right)}\left(x_{i}+1\right)=\sum_{\varepsilon_{i}\in \Lambda }B_{n-1}^{\left(\varepsilon_{i}\right)}\Phi_{n-1}^{\left(i\right)}\left(x_{i}-\varepsilon_{i}\right),\;x_{i}\in Z,$$
where Λ={−1,+1} as specified in Section 3. Thus, by taking tensor product, we get
$$\overset{d}{\underset{i=1}{⨂}}\Phi_{n}^{\left(i\right)}\left(x_{i}\right)=\sum_{\varepsilon \in \Lambda^{d}}\left(\overset{d}{\underset{i=1}{⨂}}B_{n-1}^{\left(\varepsilon_{i}\right)}\right)\left(\overset{d}{\underset{i=1}{⨂}}\Phi_{n-1}^{\left(i\right)}\left(x_{i}-\varepsilon_{i}\right)\right),\;x=\left(x_{1},x_{2},\cdots ,x_{d}\right)\in Z^{d},\;n\geq 1,$$
where $\varepsilon =(\varepsilon_{1},\varepsilon_{2},\cdots ,\varepsilon_{d})$. Taking the action of operator K on both sides and then using (Equation 15) yields
$$K\left(\overset{d}{\underset{i=1}{⨂}}\Phi_{n}^{\left(i\right)}\left(x_{i}\right)\right)=\sum_{\varepsilon \in \Lambda^{d}}K\left(\overset{d}{\underset{i=1}{⨂}}B_{n-1}^{\left(\varepsilon_{i}\right)}\right)\left(\overset{d}{\underset{i=1}{⨂}}\Phi_{n-1}^{\left(i\right)}\left(x_{i}-\varepsilon_{i}\right)\right)=\sum_{\varepsilon \in \Lambda^{d}}C_{n-1}^{\left(\varepsilon \right)}K\left(\overset{d}{\underset{i=1}{⨂}}\Phi_{n-1}^{\left(i\right)}\left(x_{i}-\varepsilon_{i}\right)\right),$$
$x=\left(x_{1},x_{2},\cdots ,x_{d}\right)\in Z^{d}$,n≥1, which together with (Equation 34) and Theorem 3 implies that
(35)$$W_{n}^{\prime }\left(x\right)=\sum_{\varepsilon \in \Lambda^{d}}C_{n-1}^{\left(\varepsilon \right)}W_{n-1}^{\prime }\left(x-\varepsilon \right)=\left(U_{n-1}^{\left(d\right)}W_{n-1}^{\prime }\right)\left(x\right),$$
$x=\left(x_{1},x_{2},\cdots ,x_{d}\right)\in Z^{d}$,n≥1. Thus
$$W_{n}^{\prime }=U_{n-1}^{\left(d\right)}W_{n-1}^{\prime }=\left(\prod_{k=0}^{n-1}U_{k}^{\left(d\right)}\right)W_{0}^{\prime },\;n\geq 1,$$
which, together with the fact $W_{0}^{\prime }=W_{0}$ and Theorem 4, implies that $W_{n}^{\prime }=W_{n}$, which together with (Equation 34) gives (Equation 32). This compete the proof. □

**Remark** **2.**
*According to [9], the unitary operators $U_{n}$, n≥0 described in Lemma 4 serve as the evolution operators of the quantum walk introduced in [9], namely the one-dimensional QBN walk. Thus, the sequence ${\left(\Phi_{n}^{\left(i\right)}\right)}_{n\geq 0}$ in Theorem 6 is exactly the state sequence of the one-dimensional QBN walk corresponding to the initial state $\Phi_{0}^{\left(i\right)}=\varphi^{\left(i\right)}$. Formula (Equation 32) then gives a link between amplitudes of the d-dimensional QBN walk and those of the one-dimensional QBN walk.*


As an immediate consequence of Theorem 6, we have the following useful corollary, which offers a formula for calculating the probability to find out the walker at a position in $Z^{d}$.

**Corollary** **1.**
*Let $\varphi^{\left(1\right)}$, $\varphi^{\left(2\right)}$, ⋯, $\varphi^{\left(d\right)}\in l^{2}\left(Z,H\right)$ be unit vectors in $l^{2}\left(Z,H\right)$. Suppose that the initial state $W_{0}$ of the d-dimensional QBN walk takes the following form*
(36)$$W_{0}\left(x\right)=K\left(\overset{d}{\underset{i=1}{⨂}}\varphi^{\left(i\right)}\left(x_{i}\right)\right),\;x=\left(x_{1},x_{2},\cdots ,x_{d}\right)\in Z^{d}.$$

*Then, for all n≥1, the state $W_{n}$ of the d-dimensional QBN walk at time n satisfies that*
(37)$$∥W_{n}\left(x\right){∥}^{2}=\prod_{i=1}^{d}∥\Phi_{n}^{\left(i\right)}\left(x_{i}\right){∥}^{2},\;x=\left(x_{1},x_{2},\cdots ,x_{d}\right)\in Z^{d},$$
*where*
(38)$$\Phi_{n}^{\left(i\right)}=\left(\prod_{k=0}^{n-1}U_{k}\right)\Phi_{0}^{\left(i\right)}$$
*with $\Phi_{0}^{\left(i\right)}=\varphi^{\left(i\right)}$ for all index i∈{1,2,⋯,d}.*


### 4.2. Limit Probability Distribution

For k≥0, we write $\Xi_{k}=\partial_{k}^{*}+\partial_{k}$, where $\partial_{k}^{*}$ and $\partial_{k}$ are the creation and annihilation operators on H, see Section 2 for details. By Lemma 2, $\Xi_{k}$, k≥0 make a commutative sequence of self-adjoint operators on H. Moreover, by the CAR in equal time, one has
$$\Xi_{k}^{2}={\left(\partial_{k}^{*}+\partial_{k}\right)}^{2}=\partial_{k}^{*}\partial_{k}+\partial_{k}\partial_{k}^{*}=I,\;k\geq 0,$$
where *I* denotes the identity operator on H. In the following, we write $\Xi_{\emptyset }=I$ and
(39)$$\Xi_{\tau }=\prod_{k\in \tau }\Xi_{k},\;\tau \neq \emptyset ,\;\tau \in \Gamma .$$

It can be verified that $\left\{\Xi_{\tau }|\tau \in \Gamma \right\}$ form a commutative family of self-adjoint unitary operators on H.

For n≥0, we write $N_{n}=\left\{0,1,\cdots ,n\right\}$. Additionally, for n≥1 and $j\in N_{n}$, we define a functional $f{\;}_{nj}$ on H in the following manner
(40)fnj(ξ)=∑σ∈▵jn−1∥∑τ⊂Nn−1(−1)#(σ∖τ)Ξτξ∥2−2nnj,ξ∈H,
where ${▵}_{j}^{n-1}=\left\{\sigma |\#\sigma =j,\;\sigma \subset N_{n-1}\right\}$ and #σ means the cardinality of σ as a set.

**Theorem** **7.**
*Let ${\left(\Phi_{n}\right)}_{n\geq 0}$ be a sequence of unit vectors in $l^{2}\left(Z,H\right)$ satisfying*
(41)$$\Phi_{n}=\left(\prod_{k=0}^{n-1}U_{k}\right)\Phi_{0},\;n\geq 1.$$
*Suppose that $\Phi_{0}$ is localized, namely $\Phi_{0}$ satisfies the following requirement*(42)$$\Phi_{0}\left(x\right)=\left\{\begin{array}{cc} \xi , & x=0;\\ 0, & x\neq 0,\;x\in Z, \end{array}\right.$$*where ξ∈H is a unit vector. Then, for all n≥1, it holds that*(43)∥Φn(x)∥2=14nfnj(ξ)+12nnj,x=n−2j,j∈Nn;0,otherwise. □

**Proof.** For each n≥1, by using the method of Fourier transform for vector-valued functions, we can get an expression of $\Phi_{n}$ of the following form
(44)$$\Phi_{n}\left(x\right)=\left\{\begin{array}{cc} \sum_{\sigma \in {▵}_{j}^{n-1}}L_{\sigma }R_{N_{n-1}\setminus \sigma }\xi , & x=n-2j,\;j\in N_{n};\\ 0, & \mathrm{otherwise}. \end{array}\right.$$Let n≥1 and $\sigma \subset N_{n-1}$. Then, with the notation $\varepsilon \left(k\right)=1-21_{\sigma }\left(k\right)$, we have
$$\begin{array}{cc} L_{\sigma }R_{N_{n-1}\setminus \sigma }\; & =\prod_{k\in \sigma }\left[\frac{1}{2}\left(\Xi_{k}-I\right)\right]\prod_{k\in N_{n-1}\setminus \sigma }\left[\frac{1}{2}\left(\Xi_{k}+I\right)\right]\\ \; & =\frac{1}{2^{n}}\prod_{k\in \sigma }\left(\Xi_{k}+\varepsilon \left(k\right)I\right)\prod_{k\in N_{n-1}\setminus \sigma }\left(\Xi_{k}+\varepsilon \left(k\right)I\right)\\ \; & =\frac{1}{2^{n}}\prod_{k=0}^{n-1}\left(\Xi_{k}+\varepsilon \left(k\right)I\right)\\ \; & =\frac{1}{2^{n}}\sum_{\tau \subset N_{n-1}}\left[\prod_{k\in N_{n-1}\setminus \tau }\varepsilon \left(k\right)\right]\;\Xi_{\tau }. \end{array}$$On the other hand, for each $\tau \subset N_{n-1}$, it follows easily that
$$\prod_{k\in N_{n-1}\setminus \tau }\varepsilon \left(k\right)=\prod_{k\in (N_{n-1}\setminus \tau )\cap \sigma }\left(-1\right)={\left(-1\right)}^{\#(\sigma \setminus \tau )}.$$Thus
$$L_{\sigma }R_{N_{n-1}\setminus \sigma }=\frac{1}{2^{n}}\sum_{\tau \subset N_{n-1}}\left[\prod_{k\in N_{m}\setminus \tau }\varepsilon \left(k\right)\right]\;\Xi_{\tau }=\frac{1}{2^{n}}\sum_{\tau \subset N_{n-1}}{\left(-1\right)}^{\#(\sigma \setminus \tau )}\;\Xi_{\tau },$$
which implies that
(45)$$∥L_{\sigma }R_{N_{n-1}\setminus \sigma }{\xi ∥}^{2}=\frac{1}{4^{n}}∥\sum_{\tau \subset N_{n-1}}{\left(-1\right)}^{\#(\sigma \setminus \tau )}\;\Xi_{\tau }\xi {∥}^{2}.$$Note that vectors $L_{\sigma }R_{N_{n-1}\setminus \sigma }\xi$ and $L_{\tau }R_{N_{n-1}\setminus \tau }\xi$ are orthogonal for σ, $\tau \in {▵}_{j}^{n-1}$ with σ≠τ. Thus, by (Equation 44) and (Equation 45), we have
(46)$$∥\Phi_{n}{\left(x\right)∥}^{2}=\left\{\begin{array}{cc} \frac{1}{4^{n}}\sum_{\sigma \in {▵}_{j}^{n-1}}∥\sum_{\tau \subset N_{n-1}}{\left(-1\right)}^{\#(\sigma \setminus \tau )}\;\Xi_{\tau }\xi {∥}^{2},\; & x=n-2j,\;j\in N_{n};\\ 0, & \mathrm{otherwise}, \end{array}\right.$$
which together with the definition of functional $f_{nj}$ gives (Equation 43). □

**Definition** **3.**
*A vector ξ∈H is said to have the ABD property if there exist constants c≥0 and r>1 such that*
(47)$$|f{\;}_{nj}\left(\xi \right)|\leq \frac{c4^{n}}{{\left(n+j\right)}^{r}},\;\forall \;j\in N_{n},\;\forall \;n\geq 1.$$


**Example** **1.**
*Every basis vector in the canonical OBN $Z=\{Z_{\sigma }|\sigma \in \Gamma \}$ of H has the ABD property.*


**Proof.** Let σ∈Γ. Then, for all τ∈Γ, we have $\Xi_{\tau }Z_{\sigma }=Z_{\sigma △\tau }$. On the other hand, we can verify that $\left\{Z_{\sigma △\tau }|\tau \in \Gamma \right\}$ make an orthonormal system in H. Thus, for any n≥1 and $j\in N_{n}$, we have
fnj(Zσ)=∑γ∈▵jn−1∥∑τ⊂Nn−1(−1)#(γ∖τ)Zσ△τ∥2−2nnj=∑γ∈▵jn−1∑τ⊂Nn−11−2nnj=∑γ∈▵jn−12n−2nnj=0,
which implies that $Z_{\sigma }$ has the ABD property. □

**Theorem** **8.**
*Let ξ∈H be a unit vector having the ABD property and ${\left(\Phi_{n}\right)}_{n\geq 0}$ a sequence of unit vectors in $l^{2}\left(Z,H\right)$ satisfying*
(48)$$\Phi_{n}=\left(\prod_{k=0}^{n-1}U_{k}\right)\Phi_{0},\;n\geq 1.$$

*Suppose that $\Phi_{0}$ satisfies the requirement below*
(49)$$\Phi_{0}\left(x\right)=\left\{\begin{array}{cc} \xi , & x=0;\\ 0, & x\neq 0,\;x\in Z. \end{array}\right.$$

*Then it holds that*
(50)$$\lim_{n\to \infty }\sum_{x\in Z}e^{\frac{itx}{\sqrt{n}}}{∥\Phi_{n}\left(x\right)∥}^{2}=e^{-\frac{t^{2}}{2}},\;t\in R.$$


**Proof.** Let t∈R. Then, by Theorem 7, we have
$$\sum_{x\in Z}e^{\frac{itx}{\sqrt{n}}}{∥\Phi_{n}\left(x\right)∥}^{2}=\frac{1}{4^{n}}\sum_{j=0}^{n}e^{\frac{it(n-2j)}{\sqrt{n}}}f_{nj}\left(\xi \right)+\cos^{n}\frac{t}{\sqrt{n}},\;n\geq 1.$$On the other hand, since ξ has the ABD property, there exist constant c≥0 and r>1 such that
$$|f{\;}_{nj}\left(\xi \right)|\leq \frac{c4^{n}}{{\left(n+j\right)}^{r}},\;\forall \;j\in N_{n},\;\forall \;n\geq 1,$$
which implies that
$$|\frac{1}{4^{n}}\sum_{j=0}^{n}e^{\frac{it(n-2j)}{\sqrt{n}}}f_{nj}\left(\xi \right)|\leq c\sum_{j=0}^{n}\frac{1}{{\left(n+j\right)}^{r}}\leq c\sum_{j=0}^{\infty }\frac{1}{{\left(n+j\right)}^{r}},\;n\geq 1,$$
which, together with $\lim_{n\to \infty }\sum_{j=0}^{\infty }\frac{1}{{\left(n+j\right)}^{r}}=0$, yields that
$$\lim_{n\to \infty }\frac{1}{4^{n}}\sum_{j=0}^{n}e^{\frac{it(n-2j)}{\sqrt{n}}}f_{nj}\left(\xi \right)=0.$$Therefore
$$\lim_{n\to \infty }\sum_{x\in Z}e^{\frac{itx}{\sqrt{n}}}{∥\Phi_{n}\left(x\right)∥}^{2}=\lim_{n\to \infty }\frac{1}{4^{n}}\sum_{j=0}^{n}e^{\frac{it(n-2j)}{\sqrt{n}}}f_{nj}\left(\xi \right)+\lim_{n\to \infty }\cos^{n}\frac{t}{\sqrt{n}}=e^{-\frac{t^{2}}{2}}.$$This completes the proof. □

The next result establishes a limit theorem for the *d*-dimensional QBN walk, which shows that for a range of choices of its initial state the *d*-dimensional QBN walk has a limit probability distribution of *d*-dimensional standard Gauss type.

**Theorem** **9.**
*Let the initial state $W_{0}$ of the d-dimensional QBN walk take the following form*
(51)$$W_{0}\left(x\right)=\left\{\begin{array}{cc} K\left(\overset{d}{\underset{i=1}{⨂}}\xi^{\left(i\right)}\right), & x=(0,0,\cdots ,0);\\ 0, & x\neq (0,0,\cdots ,0),\;\;x\in Z^{d}, \end{array}\right.$$
*where $\xi^{\left(1\right)}$, $\xi^{\left(2\right)}$, ⋯, $\xi^{\left(d\right)}\in H$ are unit vectors. For n≥1, let $X_{n}$ be a d-dimensional random vector with the probability distribution given by*
(52)$$P\left\{X_{n}=x\right\}={∥W_{n}\left(x\right)∥}^{2},\;x=\left(x_{1},x_{2},\cdots ,x_{d}\right)\in Z^{d},$$
*where $W_{n}$ is the state of the d-dimensional QBN walk at time n. Suppose that all the above vectors $\xi^{\left(1\right)}$, $\xi^{\left(2\right)}$, ⋯, $\xi^{\left(d\right)}$ have the ABD property. Then*
(53)$$\frac{X_{n}}{\sqrt{n}}⟹N\left(0,I_{d\times d}\right),$$
*namely $\frac{X_{n}}{\sqrt{n}}$ converges in law to the d-dimensional standard Gaussian distribution $N(0,I_{d\times d})$ as n→∞.*


**Proof.** For each i∈{1,2,⋯,d}, we can define a function $\varphi^{\left(i\right)}:Z\to H$ in the following manner:
$$\varphi^{\left(i\right)}\left(x\right)=\left\{\begin{array}{cc} \xi^{\left(i\right)}, & x=0;\\ 0, & x\neq 0,\;x\in Z. \end{array}\right.$$Clearly, $\varphi^{\left(1\right)}$, $\varphi^{\left(2\right)}$, ⋯, $\varphi^{\left(d\right)}$ are unit vectors in $l^{2}\left(Z,H\right)$, and moreover they admit the following relations
$$W_{0}\left(x\right)=K\left(\overset{d}{\underset{i=1}{⨂}}\varphi^{\left(i\right)}\left(x_{i}\right)\right),\;x=\left(x_{1},x_{2},\cdots ,x_{d}\right)\in Z^{d}.$$Thus, by Corollary 1, we have
(54)$$∥W_{n}\left(x\right){∥}^{2}=\prod_{i=1}^{d}∥\Phi_{n}^{\left(i\right)}\left(x_{i}\right){∥}^{2},\;x=\left(x_{1},x_{2},\cdots ,x_{d}\right)\in Z^{d},$$
where
(55)$$\Phi_{n}^{\left(i\right)}=\left(\prod_{k=0}^{n-1}U_{k}\right)\Phi_{0}^{\left(i\right)}$$
with $\Phi_{0}^{\left(i\right)}=\varphi^{\left(i\right)}$ for all index i∈{1,2,⋯,d}.Now, let us consider the characteristic function $C_{\frac{X_{n}}{\sqrt{n}}}\left(t\right)$ of random vector $\frac{X_{n}}{\sqrt{n}}$. By definition, we have
(56)$$C_{\frac{X_{n}}{\sqrt{n}}}\left(t\right)=\sum_{x\in Z^{d}}e^{\frac{i}{\sqrt{n}}\sum_{i=1}^{d}t_{i}x_{i}}∥W_{n}\left(x\right){∥}^{2},\;t=\left(t_{1},t_{2},\cdots ,t_{d}\right)\in R^{d},$$
where $x=(x_{1},x_{2},\cdots ,x_{d})$. Using (Equation 54) gives
(57)$$C_{\frac{X_{n}}{\sqrt{n}}}\left(t\right)=\sum_{x\in Z^{d}}\prod_{i=0}^{d}e^{\frac{it_{i}x_{i}}{\sqrt{n}}}∥\Phi_{n}^{\left(i\right)}\left(x_{i}\right){∥}^{2}=\prod_{i=0}^{d}\sum_{x_{i}\in Z}e^{\frac{it_{i}x_{i}}{\sqrt{n}}}∥\Phi_{n}^{\left(i\right)}\left(x_{i}\right){∥}^{2},$$
$t=\left(t_{1},t_{2},\cdots ,t_{d}\right)\in R^{d}$. For each i∈{1,2,⋯,d}, by using Theorem 8, we find
$$\lim_{n\to \infty }\sum_{x_{i}\in Z}e^{\frac{it_{i}x_{i}}{\sqrt{n}}}∥\Phi_{n}^{\left(i\right)}\left(x_{i}\right){∥}^{2}=e^{-\frac{t_{i}^{2}}{2}},\;t_{i}\in R.$$Therefore
$$\lim_{n\to \infty }C_{\frac{X_{n}}{\sqrt{n}}}\left(t\right)=\prod_{i=0}^{d}e^{-\frac{t_{i}^{2}}{2}}=e^{-\frac{1}{2}\sum_{i=1}^{d}t_{i}^{2}},\;t=\left(t_{1},t_{2},\cdots ,t_{d}\right)\in R^{d},$$
which implies that $\frac{X_{n}}{\sqrt{n}}$ converges in law to the *d*-dimensional standard Gaussian distribution. □

## 5. Conclusions Remarks

As is well known, the Hadamard walk is a one-dimensional quantum walk, whose coin space is a two-dimensional space (typically $C^{2}$). In 2002, by extending the Hadamard walk to a higher dimensional case, Mackay et al. [11] actually introduced a *d*-dimensional quantum walk for a general d≥2. However, their *d*-dimensional quantum walk takes a $2^{d}$-dimensional space as its coin space, hence has a finite number of internal degrees of freedom. In other words, as a higher dimensional extension of the Hadamard walk, the *d*-dimensional quantum walk introduced by Mackay et al. [11] does not share the same coin space with the Hadamard walk.

As is seen, in this paper we introduce a *d*-dimensional quantum walk in terms of quantum Bernoulli noises, which is called the *d*-dimensional QBN walk. The coin space of the *d*-dimensional QBN walk is the space H of square integrable Bernoulli functionals, which is infinite-dimensional. Thus the *d*-dimensional QBN walk has infinitely many internal degrees of freedom. Moreover, the *d*-dimensional QBN walk shares the same coin space H with the one-dimensional QBN walk (namely the one recently introduced in [9]), although it is a higher dimensional extension of the latter.

It should be noted that the existence of a unitary isomorphism $K:H^{⊗d}\to H$ plays a key role in constructing the *d*-dimensional QBN walk. For a finite dimensional space, say $C^{2}$, there exists no unitary isomorphism from ${\left(C^{2}\right)}^{⊗d}$ to $C^{2}$ unless d=1. This just means that our approach in this paper differs from that used by Mackay et al. in [11].

Decoherence is one of important topics in the study of quantum walks. Physically, decoherence means a deviation from pure quantum behavior. If a quantum walk shows some classical asymptotic behavior, then it contains an amount of decoherence. Kendon and Tregenna [16] showed for the first time that decoherence can be useful in quantum walks. Brun et al. [17] investigated quantum walks with decoherent coins. Chisaki et al. [18] analyzed a class of quantum walks with position measurements and found that those walks have limit probability distributions of Gauss type under some situations, which means that quantum walks with position measurements can produce decoherence. There are other works addressing decoherence in quantum walks (see [10] and references therein). As is seen, as a model of higher-dimensional quantum walk constructed in terms of quantum Bernoulli noises, the *d*-dimensional QBN walk has a limit probability distribution of *d*-dimensional standard Gauss type for some choices of its initial state, which together with the work of [9] implies that quantum Bernoulli noises can provide an alternative way to produce decoherence in quantum walks.

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
