# Peer review of "Higher-Dimensional Quantum Walk in Terms of Quantum Bernoulli Noises"

_entropy, 2020, doi:10.3390/e22050504_

Round 1

Reviewer 1 Report

The authors studied the discrete-time quantum walk with high dimensional coin under the noise. The mathematical exact result is not surprising but is rigorous to be derived. Therefore, this analytical result is worth for being published from Entropy. I think that the related paper [International Journal of Quantum Information, Vol.6, Issue 6, pp.1231-1243 (2008)] should be commented. On the noise, the related analysis [Quantum Information and Computation 11 (2011) pp.0741-0760] can be also commented. These results are slightly non-intuitive since the variance of the Gaussian distribution is different from the standard case. Is there the discontinuity of your model? If not, the authors should comment why the discontinuity did not see.

Reviewer 2 Report

In this manuscript, the authors present an extension of the quantum walk in terms of quantum Bernoulli noises [15] to a higher dimensional lattice. They also prove that, for a range of initial states, the walk has a limit probability distribution of d-dimensional Gauss type.

I must admit that this paper, being mathematically oriented, lies a bit far from my field of expertise. However, after a careful reading of the manuscript, and consultation of references [13], [15], and Quantum Inf Process 17, 46 (2018), I am convinced about the robustness and rigor of the results presented in the manuscript. I therefore recommend publication. I would like to strongly encourage the authors to make an effort to present their work to physicist working on quantum walks in a more pedagogical way in future papers, and to discuss with them about the possibility to implement the QBN walk using some experimental setup. This experimental side, along with its theoretical understanding, constitutes the basis of the growth of the research in quantum walks. Some questions remain to be investigated. In particular, the appearance of the Gaussian probability is well understood in the standard QW. Is a similar mechanism acting for the QBN walk? Is there a continuum limit, as in https://arxiv.org/abs/1911.09791, that could provide more insight into the process? In my opinion, these questions and comments deserve further research.

Round 2

Reviewer 1 Report

The authors reply satisfied to recommend to be published as the present form.